# Dual Biologic Therapy in Moderate to Severe Pediatric Inflammatory Bowel Disease: A Retrospective Study

**DOI:** 10.3390/children10010011

**Published:** 2022-12-21

**Authors:** Magdalena Wlazło, Monika Meglicka, Anna Wiernicka, Marcin Osiecki, Jarosław Kierkuś

**Affiliations:** Department of Gastroenterology, Hepatology, Feeding Disorders and Pediatrics, The Children’s Memorial Health Institute, 04-730 Warsaw, Poland

**Keywords:** pediatric gastroenterology, dual biologic, Crohn’s disease, ulcerative colitis

## Abstract

Background: Inflammatory bowel diseases in children are characterized by a wide variety of symptoms and often a severe clinical course. In the treatment, we aimed to induce and maintain remission. We focused on assessing the efficacy and safety of the concomitant use of two biologic therapies including: anti-TNF (infliximab, adalimumab) vedolizumab and ustekinumab in a refractory pediatric IBD cohort. Methods: Fourteen children (nine ulcerative colitis, one ulcerative colitis/IBD-unspecified, four Crohn’s disease) with a disease duration of 5.2 (8 months–14 years) years, initiated dual therapy at an age of 11.7 (3–17) years after failure of monotherapy with a biological drug. Five patients (36%) were treated with vedolizumab/adalimumab (VDZ + ADA), five (36%) with ustekinumab/adalimumab (UST + ADA), and three (21%) with infliximab/vedolizumab (IFX + VDZ). One patient (7%) was switched from a combination of vedolizumab and adalimumab to ustekinumab and adalimumab during follow-up. Results: A clinical improvement was obtained in ten children (73%; 5 UC, 1 UC/IBD-unspecified, 4 CD) on the PCDAI/PUCAI scale after 4 months of a second biological drug being added. The median fecal calprotectin decreased from 1610 µg/g (140–10,100) to 586 µg/g (5–3410; *p* = 0.028) between baseline and 4 months. Conclusions: Our clinical experience suggests that dual therapy may be an option for pediatric patients with moderate and severe courses of IBD with limited therapeutic options

## 1. Introduction

Inflammatory bowel disease (IBD) includes a group of chronic diseases (Crohn’s disease, ulcerative colitis, and unclassified IBD) of the gastrointestinal tract. The manifestation of these diseases is the result of complex interactions among the environment, genetic predisposition, immunological system, and microbial flora [1,2]. Clinical manifestations are usually more severe in pediatric IBD, especially VEO-IBD, than in adult IBD patients [3].

Children are more likely to present pancolitis and are less effected by steroid treatment than adults. According to published data, 40% of pediatric patients require colectomy within 10 years of diagnosis. In comparison, this problem affects only 20% of adults UC patients [4,5]. Other publications estimated the need for colectomy in children with UC at 20% within 5 years of the onset [6].

The primary therapeutic goal is to induce and maintain remission. In the pediatric population, treatment with corticosteroids and immunomodulatory drugs may cause long-term side effects such as growth failure, osteopenia, and pathological fractures [7,8]. Corticosteroid resistance or dependence is common in pediatric patients [9].

For this reason, biological therapies are now commonly used in patients with moderate-to-severe early IBD, but only have remission rates of approximately 40% after one year [10]. Most treatment experiences for Crohn’s disease (CD) and ulcerative colitis (UC) consist of a combination of targeted biological drugs with immunomodulators [11]; however, there is limited literature on the effects of long-term combination biological therapies in IBD.

We aimed to assess the efficacy and safety of two biological therapies, anti-TNF (infliximab, adalimumab) vedolizumab and ustekinumab, in a refractory pediatric IBD cohort. The indicated group of biological drugs, apart from anti-TNF inhibitors (infliximab, adalimumab), is used off-label in pediatric patients.

## 2. Materials and Methods

We present our experience with combining dual biologic therapy in PIBD between June 2021 and April 2022 in our Department of Gastroenterology, Hepatology, Feeding Disorders, and Pediatrics, The Children’s Memorial Health Institute in Warsaw. This is a retrospective study of pediatric IBD patients aged up to 18 years.

The requirement for accession to qualification for therapy was written consent signed by the legal guardian of each patient. Data for all patients undergoing therapy were retrospectively obtained from the patient’s paper and electronic medical history.

Our patients were treated with infliximab, adalimumab, vedolizumab, or ustekinumab in combination.

All patients received standard intravenous (IV) induction dosing with infliximab 5 mg/kg body weight IV, vedolizumab (for patient ≤ 30 kg–150 mg IV; ≥ 30 kg weight–300 mg IV) at day 1, week 2, week 6 and ustekinumab (for patient ≤ 55 kg–260 mg IV; 55–85 kg–390 mg IV) at day 1. The induction dosing with adalimumab for patients was (80 mg/160 mg subcutaneous (SC) injections ≥ 40 kg weight) on day 1, followed by 40 mg/80 mg on week 2. The maintenance dose was 40 mg every 2 weeks. Maintenance dosing regimens for infliximab was 5 mg/kg IV, vedolizumab was either 150 mg or 300 mg iv (≥30 kg body weight) IV, ustekinumab 45 mg/90 mg (≥40 kg body weight) SC injections, every 8 weeks for all medications.

We reviewed each patient for disease characteristics, medical treatment history, including the age of onset and the duration of the disease, oral steroid and immunomodulator use, previous biologics and other medications, adverse events, and surgeries, which were recorded. Disease activity (PUCAI and PCDAI scale), fecal calprotectin, C-reactive protein (CRP), erythrocyte sedimentation rate (ESR), and albumin levels were analyzed. (Table 1 and Table 2) PUCAI and PCDAI indices were used to assess the clinical condition of the patients. The total PUCAI score (range 0–85 points) and total PCDAI score (range 0–100 points), respectively.

This retrospective study involved 14 patients; however, due to the fact that one of them changed treatment schedule during the 4-month (+/−2 weeks) follow-up, 15 treatment schedules were used for statistical calculations.

We searched for laboratory results and clinical activity scores at baseline and at least four months (+/−2 weeks) after the initiation of dual biological therapy.

### Disease Activity

Primary outcome was clinical response defined as a decrease in PCDAI of at least 12.5 points between baseline and 4 months after the dose of a second biological drug for CD and a decrease in PUCAI of at least 20 points between baseline and this time for UC/IBD-U. Secondary outcomes included clinical remission defined as having a PCDAI score ≤ 10 points and PUCAI score ≤ 10 points between baseline and 4 months, adverse events and changes in serum biomarkers (CRP and ESR), fecal calprotectin, and albumin levels as nutritional status between baseline and 4 months.

Fecal calprotectin improvement was defined as a 50% decrease within 4 months of follow-up.

Additionally, six patients diagnosed with UC, have the results of endoscopic examinations 4 months after the start of therapy.

Endoscopic remission was defined as an endoscopic Mayo score ≤ 1 for UC, and endoscopic response was defined as a decrease of >1 in the endoscopic Mayo score.

## 3. Results

Fourteen children (nine with ulcerative colitis, one with ulcerative colitis/IBD-U, four with Crohn’s disease) with a disease duration of 5.2 years (8 months–14 years) initiated dual therapy at the age of 11.7 (3–17) years after failure of monotherapy with a biological drug. Five patients (36%) were treated with vedolizumab/adalimumab (VDZ + ADA), five (36%) with ustekinumab/adalimumab (UST + ADA), and 3 (21%) with infliximab/vedolizumab (IFX + VDZ). One patient (7%) initially treated with dual biological therapy in the regimen (VDZ + ADA), converted to treatment (UST + ADA) after 4 months of ineffective treatment. For this patient, there was significant improvement in the following months of observation

### 3.1. Clinical Outcomes

Clinical improvement was obtained in ten children (73%; 5 UC, 1 UC/IBD-U, and 4 CD) on the PCDAI/PUCAI scale after 4 months of a second biological drug dose. One patient improved significantly after 4 months of follow-up but required a colectomy two months later (UC, IFX + VDZ). Two patients in this group (14%; UC, VDZ + ADA; UC, IFX + VDZ) discontinued treatment prematurely [see: Section 3.2].

The median PCDAI decreased from 52.25 points (35–65.5) to 10 points (5–15; *p* = 0.067) in four patients with Crohn’s disease. The median PUCAI reduced from 40 points (0–85) to 5 points (0–45; *p* = 0.024) between baseline and 4 months in seven patients diagnosed with ulcerative colitis, who were treated for a minimum of 4 months (Tabel 2)

One patient with an unclassified form of inflammatory bowel disease with a baseline diagnosis of ulcerative colitis was scored on the PUCAI scale. On the day of enrollment in the treatment with dual biologic therapy, the patient was assessed using the PUCAI scale at 85 points, and after 4 months of treatment with (VDZ + ADA) she obtained a score of 45. Owing to the unsatisfactory effect, the treatment regimen was changed to (UST + ADA) with a significant improvement and a PUCAI score of 25 points in the fourth month of therapy.

Seven (47%) patients including five (45%) with UC or IBD-unspecified and two with (50%) CD achieved remission at 4 months.

We compared the fecal calprotectin in all cases that completed the 4-month follow-up (87%). The median fecal calprotectin decreased significantly from 1610 µg/g (140–10,100) to 586 µg/g (5–3410 *p* = 0.028) between baseline and 4 months. Two patients (one UC and one CD) showed an increase in calprotectin levels. In case of a patient with an unclassified form of IBD, the concentration of calprotectin increased after the first four months of therapy, followed by a marked drop after changing the treatment regimen. This was accompanied by improvement in the patient’s clinical condition.

The median CRP reduced from 0.5 mg/dL (0.1–4.3) at baseline to 0.3 mg/dL (0, 1–4, 4; *p* = 0.5) after 4 months, and the median ESR decreased from 16 mm/h (2–70) to 15 mm/h (2–70; *p* = 0.24) in the same time.

The median albumin increased from 34.1 g/L (24.3–44.9) at baseline to 39.2 g/L (25.7–43.6; *p* = 0.13) at 4 months.

Six patients with ulcerative colitis (66%) underwent sigmoidoscopy at least four months after the initiation of dual biological therapy. Two patients (33%) achieved endoscopic remission, one (17%) achieved endoscopic improvement, and the other three (50%) patients had no benefit in endoscopic examination.

Most of our patients (11 (79%) of 14) were treated with at least two biologic monotherapies with no success. Some of our pediatric patients (four UC and one CD) had an increased frequency of drug dosing before the inclusion of dual therapy, especially in the case of infliximab and vedolizumab. These patients received doses every 4 weeks rather than every 8 weeks as is standard. However, in a severe exacerbation of the disease, we also decided to quickly add a faster-acting drug, such as adalimumab, to give vedolizumab time to develop fully. The majority of our patients, 13 (93%) received drugs from group immunomodulators, such as azathioprine (AZA), before starting dual biological therapy. All enrolled patients were treated with steroid therapy (100%), and on the day of qualifying for therapy, 6 (42%) of the 14 patients were receiving steroid therapy. During 4 months of treatment with dual biological therapy, steroids were successfully discontinued in four (67%) of the six patients.

Two patients (14%) presented parenteral symptoms, that is: genitourinary psoriasis lesions (CD, UST + ADA) and mouth ulcers (UC, VDZ + ADA), which resolved during the 4-month follow-up period.

One patient (7%) underwent IBD-related surgery before starting dual therapy. No serious opportunistic infections were observed in this cohort. No tumors were diagnosed. However, such a short period of patient observation does not allow the assessment of adverse events that usually occur in the long term.

### 3.2. Discontinuation of Dual Biological Therapy

Four patients (28%) showed no improvement after treatment. We report an adverse event during treatment in the form of an anal abscess. One patient showed significant improvement after 4 months of therapy but required a colectomy two months later due to exacerbation of the disease (UC, VDZ + IFX). In another patient, despite clinical improvement, dual biological therapy was discontinued because of anal abscess. (UC, VDZ + ADA). One patient had cardiac complications after suffering from a COVID-19 infection. (UC, IFX + VDZ). Lastly, a patient with UC on (VDZ + ADA) discontinued dual therapy after 4 months due to persistent symptoms, and no endoscopic improvement was achieved.

## 4. Discussion

Pediatric patients with inflammatory bowel disease under the care of our clinic, who were qualified for dual biological therapy, exhausted the available conventional therapeutic methods before starting therapy. All the children were previously hospitalized, and more than half of them were hospitalized several times before starting dual therapy.

There are limited data on the use of dual biological therapy in inflammatory bowel disease, especially in the pediatric population. Geem [12] summarized the effectiveness of monotherapy with biological drugs, indicating the need to search for new drugs or treatment regimens for children with inflammatory bowel diseases [12,13]. The chance of achieving clinical remission in patients receiving monotherapy with an anti-TNF drug is estimated to be 40–60% [14,15].

In 2020, Dolinger et al. published [16] a retrospective study of 16 pediatric patients treated with dual biological therapy. Three types of combined biological therapies were used: nine patients (56%) were treated with VDZ + tofacitinib, four (25%) with UST + VDZ, and three (19%) with UST + tofacitinib. Twelve (75%) patients achieved steroid-free clinical remission after 6 months. Two patients switched therapy with improvement. One patient required further surgery.

In another study, Olbjørn [17] used dual biological therapy consisting of infliximab and vedolizumab in eight patients with partial improvement after treatment with infliximab. Reported clinical remission was observed in four (50%) patients (three UC, one CD). Four (50%) required colectomy (three CD, one UC). Howard et al. [18] reported the use of a combination therapy with vedolizumab and ustekinumab for three IBD patients who had previously failed both monotherapies. Dual biological therapy contributed to the closure of the rectovaginal fistula in one patient. In another patient undergoing this therapy, the continuity of the gastrointestinal tract was restored after removal of the stoma.

In our study, we presented three types of procedures that combined two biological drugs. We had high hopes for the combination of vedolizumab and ustekinumab. According to the latest reports, this combination may be particularly advantageous for patients with fistulous Crohn’s disease [18,19,20]. Due to financial reasons, we were not able to start such treatment in our patients. In the adult IBD patient population, combining a TNF-antagonist that has a rapid systemic effect with a slower intestinal specific effect an agent such as VDZ seems very attractive. This combination appears to be safe and effective [21].

The therapeutic concentration of some biological drugs, such as vedolizumab, may reach optimal levels even several months after the start of therapy. In patients who have experienced unsatisfactory improvement on vedolizumab monotherapy, the addition of an anti-TNF agent to therapy may have a bridging effect until vedolizumab becomes effective [22]. Furthermore, we avoided using of high doses of corticosteroids and the side effects associated with such therapies, which are particularly unfavorable in the pediatric population [23]. In our observations, a 50%- reduction in FC over 4 months as an indicator of improvement were used. According to the latest reports, the dynamic change of FC has great reliability in predicting an inactive disease [24].

The costs of dual therapy are higher than those of conventional treatment with immunomodulators; however, failure of standard drug therapy increases the risk of hospitalization and surgery. The evidence presented in the systematic review shows that in patients with inflammatory bowel disease after surgery, surgical complications are common and important for patients, both in terms of the cost of treatment and quality of life [25].

### Limitations

This retrospective study has some limitations. First, the short time of patient observation does not allow the establishment of a long-term safety profile of pediatric patients undergoing dual biological therapy. In addition, this was a retrospective study with a small group of 14 patients. We did not measure the serum drug or anti-drug antibody levels.

Moreover, endoscopic examinations are lacking in most patients during the first 4 months of therapy.

## 5. Conclusions

The experience of our clinic suggests that dual therapy may be an option for pediatric patients with moderate and severe courses of inflammatory bowel disease with limited therapeutic options. More randomized controlled trials are needed to compare the efficacy and safety endpoints of the combination therapies.

Before making the decision to qualify a patient for dual biological therapy, the safety and all available treatment regimens should be considered.

## Figures and Tables

**Table 1 children-10-00011-t001:** Characteristics of patients qualified for dual biological therapy-baseline day.

	Gender	Age	Disease and Location	Duration of Disease (Years)	Previous Therapy	Combinations
1	F	14	UC	2	5-ASA, GKS, IFX, VDZ	VDZ + ADA
2	M	17	UC	9	5-ASA, AZA, GKS, GOL, ETN, UST, VDZ	VDZ + ADA
3	M	16	UC	3	GKS, AZA, cyclosporine, IFX, ADA, VDZ	VDZ + ADA
4	M	3	UC	0.7	5-ASA, GKS, AZA, IFX	IFX + VDZ
5	M	3	UC	1	5-ASA, GKS, AZA, IFX	IFX + VDZ
6	M	13	UC	1	5-ASA, AZA, GKS, 2 × FMT, IFX	IFX + VDZ
7	F	5	UC	2	5-ASA, GKS, IFX, UST	UST + ADA
8	M	14	UC	14	5-ASA, AZA, GKS, IFX, VDZ, UST	UST + ADA
9	F	15	UC	11	GKS, AZA, cyclosporine, IFX, VDZ	VDZ + ADA
10	F	14	CD	5	Modulife,5-ASA, GKS, AZA, IFX, UST	UST + ADA
11	M	14	CD	1	Modulife, 5-ASA, GKS, AZA, IFX, UST	UST + ADA
12	F	16	CD	8	GKS, ADA, IFX, UST	UST + ADA
13	M	14	CD	11	GKS, AZA, IFX, VDZ	VDZ + ADA
14	F	6	IBD-unspecified	4	5-ASA, GKS, AZA, cyclosporin, ETN, IFX, VDZ	VDZ + ADA ADA + UST

ADA: adalimumab; AZA: azathioprine; CD: Crohn’s Disease; ETN: etancerept; F: female; FMT: Fecal Microbiota Transplants; GKS: glucocorticoids; GOL: golimumab; IFX: infliximab; M: male; UC: Ulcerative Colitis; UST: ustekinumab; VDZ: vedolizumab.

**Table 2 children-10-00011-t002:** Comparison characteristics of patients and response to treatment in dual biological therapy.

	Gender Age Diagnosis	Combinations	PUCAI		PCDAI		FC (µg/g)		CRP (mg/dL)		AE
			1 Day	After 4 Months	1 Day	After 4 Months	1 Day	After 4 Months	1 Day	After 4 Months	
1.	F, 14, UC	VDZ + ADA	70	x	-	-	2460	x	0.1	x	anal abscess
2.	M, 17, UC	VDZ + ADA	50	0	-	-	405	104	0.5	0.1	No
3.	M, 16, UC	VDZ + ADA	35	5	-	-	1800	127	0.4	0.1	No
4.	M, 3, UC	IFX + VDZ	75	x	-	-	4430	x	0.4	x	cardiac complications
5.	M, 3, UC	IFX + VDZ	55	0	-	-	1620	1280	0.1	0.1	Colectomy
6.	M, 13, UC	IFX + VDZ	0	5	-	-	140	415	2.2	4.4	No
7.	F, 5, UC	UST + ADA	85	0	-	-	1240	5	1.1	0.1	No
8.	M, 14, UC	UST + ADA	40	10	-	-	8230	694	0.3	0.3	No
9.	F, 15, UC	VDZ + ADA	25	45	-	-	165	55	0.1	0.2	No
10.	F, 14, CD	UST + ADA	-	-	52	7.5	2840	1590	4.3	0.9	No
11.	M, 14, CD	UST + ADA	-	-	65.5	12.5	1540	586	1.2	0.6	No
12.	F, 16, CD	UST + ADA	-	-	52.5	5	10,100	2350	1.9	3.3	No
13.	M, 14, CD	VDZ + ADA	-	-	35	15	749	988	0.6	0.9	No
14.	F, 6, IBD-unspecified	VDZ + ADA ADA + UST	85 45	45 25	-	-	2090 3410	3410 178	0.1 0.3	0.3 0.1	No

ADA: adalimumab; AE-adverse event; CD: Crohn’s Disease; CRP: C-reactive protein; F: female; FC: calprotectin; IFX: infliximab; M: male; PCDAI: Pediatric Crohn’s Disease Activity Index; PUCAI: Pediatric Ulcerative Colitis Activity Index; UC: Ulcerative Colitis; UST: ustekinumab; VDZ: vedolizumab.

## Data Availability

Data may be obtained from the corresponding author on submission of written request.

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
