# Peer review of "Dual Biologic Therapy in Moderate to Severe Pediatric Inflammatory Bowel Disease: A Retrospective Study"

_children, 2022, doi:10.3390/children10010011_

Round 1

Reviewer 1 Report

Thank you for allowing me to review this retrospective study of dual biologics in children with IBD. The authors should be commended for attempting this therapuetic approach for children with IBD, when often children are underrated with IBD therapies.

I have minor issues

1. From a methodological point of view - did you have a strict 4 month appointment? If the patient saw the clinic on day 0 then 3month then 6 months - which visit would constitute the 4 month visit?

2. Were PUCAI or Paed-CDAI performed in a standardised manner or were data from the electronic record used to complete these data points for this study>?

Results

1. How many children received dose intensified monotherapy with their biologic before attempting dual biologics? Most clinicians and tertiary centres would attempt dose intensification (either with total dose or frequency changes).

2. Do you have drug levels PRIOR to commencing dual biologic therapy? If so what are these results?

3. In the combination of VDZ and IFX did the patients have both infusions on the same day on an 8 weekly basis or in a different cycle?

4. Do the authors have comments to make about patient acceptability of this - dual biologics means more infusions or more injections - did you collect data about this aspect? 

Reviewer 2 Report

This is indeed an interesting and important topic. This paper is on a small cohort of patients, and also a retrospective study.

I lack in the introduction a presentation of the "off-label" regim - that we use several drugs not tested on children by the drugcompanies.

I also lack the information about dosing - that could be presented as meandoses in all patients or in the tables for every patient.

I also has some suggestions to make the tables better; I think all patients with UC resp CD should be presented in one table each - not like present with all patients mixed up.

I also thnk it would be better if you name the patient; pt 1, pt 2 etc in both tables so you can follow each pt;  what drugs the patients had prior dual biologics and that how it worked out for each patient in table 2.

Is this really correct: "According to published data, 40% of pediatric patients require colectomy within 10 years of diagnosis" ref 4,5 - cant find it?

Please check the english, in several places there are dangling modifiers. And the overall english; perhaps a language review could be helpful

Round 2

Reviewer 2 Report

This paper has improved after revision, and it is much easier to follow the result.